# Cell barrier function of resident peritoneal macrophages in post-operative adhesions

Tomoya Ito [1✉], Yusuke Shintani[1], Laura Fields[1], Manabu Shiraishi[1], Mihai-Nicolae Podaru[1], Satoshi Kainuma[1], Kizuku Yamashita[1], Kazuya Kobayashi[1], Mauro Perretti [1], Fiona Lewis-McDougall[1] & Ken Suzuki[1✉]

Post-operative adhesions are a leading cause of abdominal surgery-associated morbidity. Exposed fibrin clots on the damaged peritoneum, in which the mesothelial barrier is disrupted, readily adhere to surrounding tissues, resulting in adhesion formation. Here we show that resident F4/80[High]CD206[−] peritoneal macrophages promptly accumulate on the lesion and form a 'macrophage barrier' to shield fibrin clots in place of the lost mesothelium in mice. Depletion of this macrophage subset or blockage of CD11b impairs the macrophage barrier and exacerbates adhesions. The macrophage barrier is usually insufficient to fully preclude the adhesion formation; however, it could be augmented by IL-4-based treatment or adoptive transfer of this macrophage subset, resulting in robust prevention of adhesions. By contrast, monocyte-derived recruited peritoneal macrophages are not involved in the macrophage barrier. These results highlight a previously unidentified cell barrier function of a specific macrophage subset, also proposing an innovative approach to prevent post-operative adhesions.

---

[1] William Harvey Research Institute, Barts and The London School of Medicine and Dentistry, Queen Mary University of London, London, UK.
✉email: t.ito@qmul.ac.uk; ken.suzuki@qmul.ac.uk

Post-operative adhesions are pathological connections binding organs or tissues across a virtual space, e.g., the peritoneal, pericardial or pleural cavity. These are some of the leading causes of post-operative morbidity, resulting in various complications[1]. Abdominal adhesions occur in 79–93% of patients who have undergone a major abdominal or pelvic procedure[2,3]. These can cause pain and distress and may result in more critical complications, i.e., bowel obstruction and female infertility[4]. The only effective treatment for post-operative adhesions, once developed, is corrective surgery[5], whereas this procedure is invasive and often leads to a recurrence of adhesions[6], suggesting the importance of the prevention. The most common preventive approach for post-operative abdominal adhesions is currently implantation or administration of biomaterial barrier products, including artificial films, liquid, or gel, which keep the injured tissue-surface separated from the neighboring tissue/organ[7]. These barrier products are, however, effective in only half of the patients and have limitations in their practicability[8–10]. Therefore, there is a great need for a more effective and more widely applicable preventative treatment for post-operative abdominal adhesions. To this aim, it is essential to further understand the cellular and molecular mechanism underpinning the adhesion formation post-surgery.

The major initiating factor for post-operative adhesion formation is the production of fibrin clots[11]. Following a surgery-associated injury, the coagulation cascade is activated, leading to the formation of fibrin clots on the surface of damaged tissues[12]. The adhesive polymerization sites on the exposed fibrin clot promote the surrounding tissue to bind with them[13,14]. The endogenous fibrinolysis ability, in which plasmin and plasminogen activators play a role, is not so substantial as to attenuate the fibrin clot formation or adhesion formation[15]. Previous studies have suggested an involvement of various types of cells in this initial process for post-operative adhesion formation[16,17], including activated mesothelial cells and neutrophils[18–20]. However, the role of macrophages, which are a major cell type in the

peritoneal cavity, remains contentious[21–25]. For instance, clodronate liposome-induced depletion of peritoneal macrophages attenuates the intra-abdominal adhesion formation in a gauze-induced adhesion model in mice[25], suggesting a causative role of macrophages. On the other hand, adoptive transfer of peritoneal macrophages reduces the adhesion formation in a parietal peritoneal muscular defect model in rabbits[21], indicating an anti-adhesion effect of macrophages. Although the different species and/or models used in these studies may be a reason for the contradicting observations[26], the heterogeneity and diversity of peritoneal macrophages may also explain this inconsistency.

Here we show that resident F4/80$^{High}$CD206$^-$ peritoneal macrophages, but not recruited F4/80$^{Low}$CD206$^+$ macrophages, have an ability to form a "cell barrier" that determines the formation of post-operative abdominal adhesions in a mouse model. This previously unidentified macrophage barrier is formed via CD11b, but usually insufficient to fully shield the exposed fibrin clots, thereby allowing the adhesion formation. However, we describe that IL-4-based treatment augments the macrophage barrier and results in robust prevention of post-operative adhesions, proposing an innovative anti-adhesion strategy that is based on a distinctive concept from the conventional treatments.

## Results

**Abdominal adhesion formation within a day of surgery**. We first investigated the local cellular dynamics related to the abdominal adhesion formation using a mouse model based on ischemic button creation on the parietal peritoneum[27] (Supplementary Fig. 1a). The severity of adhesions was evaluated by the reported adhesion-scoring system[4,28]. This model demonstrated that the adhesions between the ischemic button and surrounding tissues were formed as early as day 1 post-surgery (Supplementary Fig. 1b, c). Histological assessments using hematoxylin and eosin staining revealed that the mesothelial layer on the ischemic button surface was disrupted by day 1 and that there was an accumulation of nucleated cells, which have different

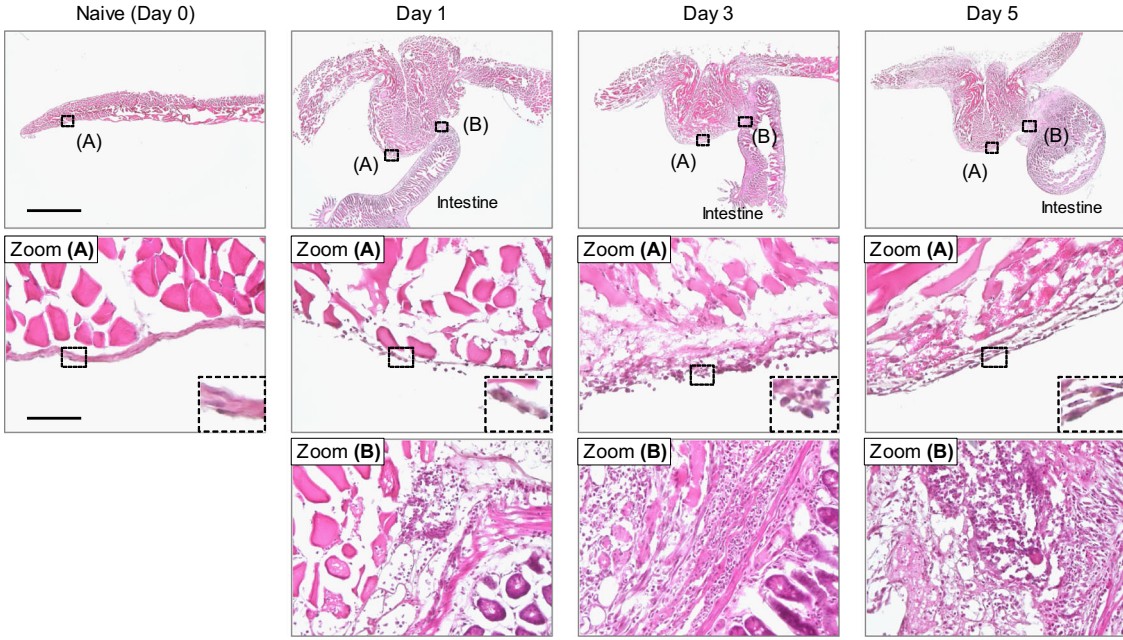

**Fig. 1 Post-operative adhesions are formed as early as 1 day after surgery in the mouse ischemic button model.** Representative images of hematoxylin-eosin staining of the ischemic button at different time points. Zooms (A) and (B) show the non-adhesion area (A) and adhesion area (B) of the ischemic button, respectively. The inserted images in Zooms (A) present higher magnification views of the black dashed boxes. The naive peritoneum was used as control (Day 0). $n = 3$ independent experiments. Scale bars represent 2 mm in the lower magnification images and 100 μm in the higher magnification images.

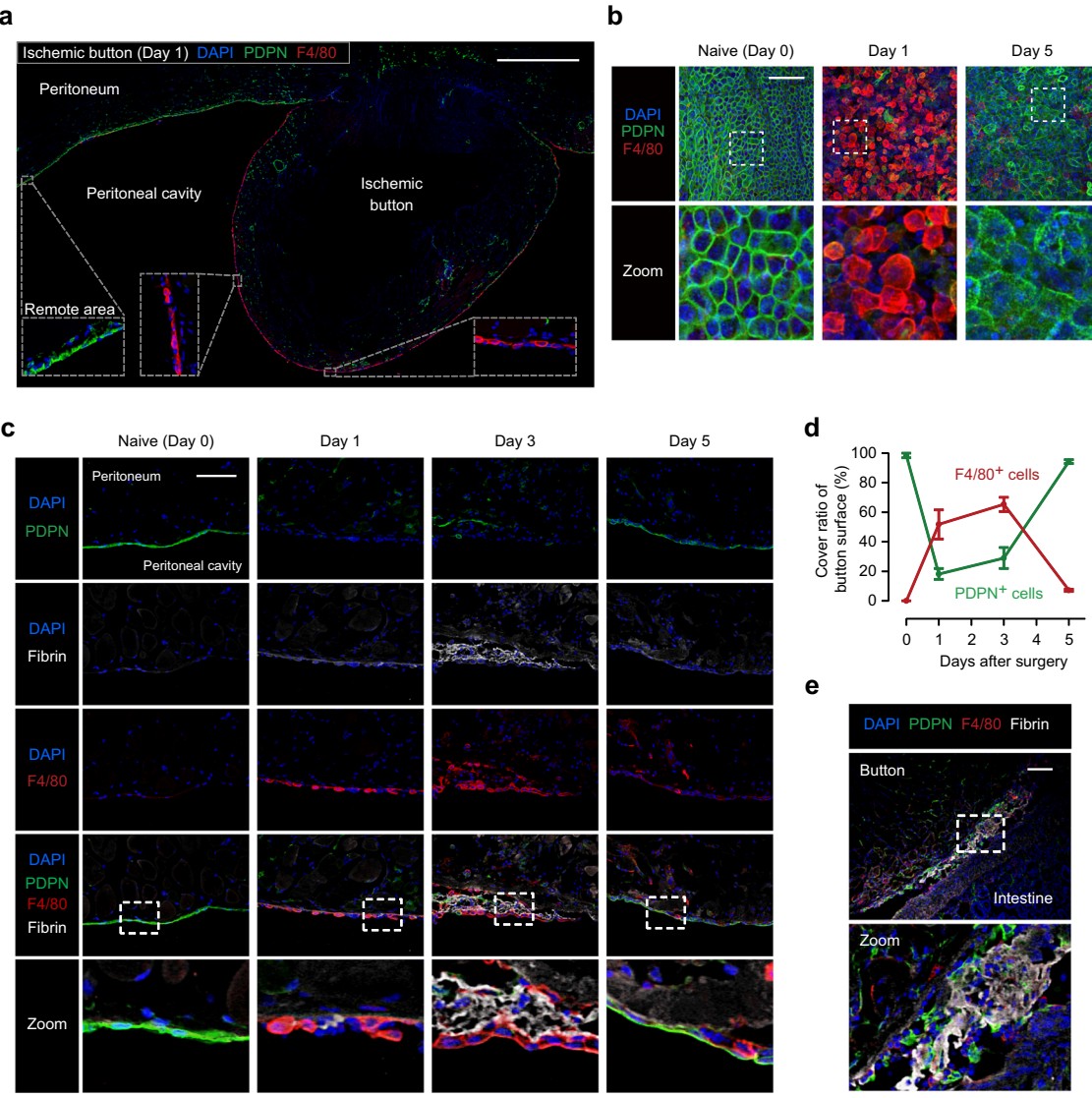

**Fig. 2 F4/80+ macrophages accumulate on fibrin clots in the damaged peritoneum. a** Representative tiling image of immunofluorescence staining for the ischemic button on day 1 post-surgery. Cross-sections were stained for PDPN and F4/80. The inserted images present the higher magnification of the surface area of the ischemic button and the remote area from the ischemic button. $n = 3$ independent ischemic buttons. Scale bar, 1 mm. **b** Whole-mount immunohistostaining of the ischemic button surface. $n = 3$ independent ischemic buttons. Scale bars, 100 μm. **c, d** Representative immunofluorescence images and quantification of ischemic buttons created in the mouse peritoneal membrane. Cross-sections were stained for PDPN, F4/80, and fibrin. The naive peritoneum was examined as "Day 0". The graph shows the time course of the cover ratio of the ischemic button surface by PDPN+ mesothelial cells or F4/80+ macrophages. $n = 6$ (Day 0), 5 (Day 1), 7 (Day 3), and 6 (Day 5) mice. Data represent the mean ± SEM. Scale bar, 100 μm. **e** Representative immunofluorescence images of the adhesion formed between the ischemic button and surrounding tissue (intestine) on day 3 post-surgery. $n = 3$ independent experiments. Scale bars, 200 μm.

morphologies from naive mesothelial cells, on the button surface (Fig. 1). At the sites of adhesion formation, the mesothelial layer was lost by day 1, and connective tissue with cell infiltration was formed to bind the ischemic button and the surrounding tissue, i.e., the intestine.

**F4/80+CD206− macrophages accumulate on fibrin clots**. To identify the accumulated cells on the ischemic button surface, we first performed immunohistofluorescent staining for F4/80 and podoplanin (PDPN) using an ischemic button sample that was free from adhesion on day 1 after surgery. The result exhibited a remarkable change in the peritoneal surface of the ischemic button. The parietal peritoneum adjacent and remote to the ischemic button was entirely shielded by the PDPN+ mesothelial cell monolayer, while by contrast the peritoneal surface of the

button was mostly covered by a monolayer of F4/80+ macrophages instead of PDPN+ cells (Fig. 2a). We then further progressed the analysis of every ischemic button by using both whole-mount and cross-section immunohistofluorescent staining. Before the surgical insult, a monolayer of PDPN+ mesothelial cells[29] with a cobblestone-like morphology completely covered the parietal peritoneum, while the majority of these PDPN+ cells disappeared from the button surface by day 1 post-surgery (Fig. 2b–d). This disappearance of mesothelial cells was further confirmed by immunostaining for two other mesothelial markers[29], cytokeratin 19 and Mesothelin (Supplementary Fig. 2a, b). The disrupted mesothelial cell layer was, however, reconstructed with PDPN+ cells by day 5 (Fig. 2b–d). It was observed that fibrin clots were formed on the surface of the ischemic button with a peak at day 3 (Fig. 2c). Of note, F4/80+

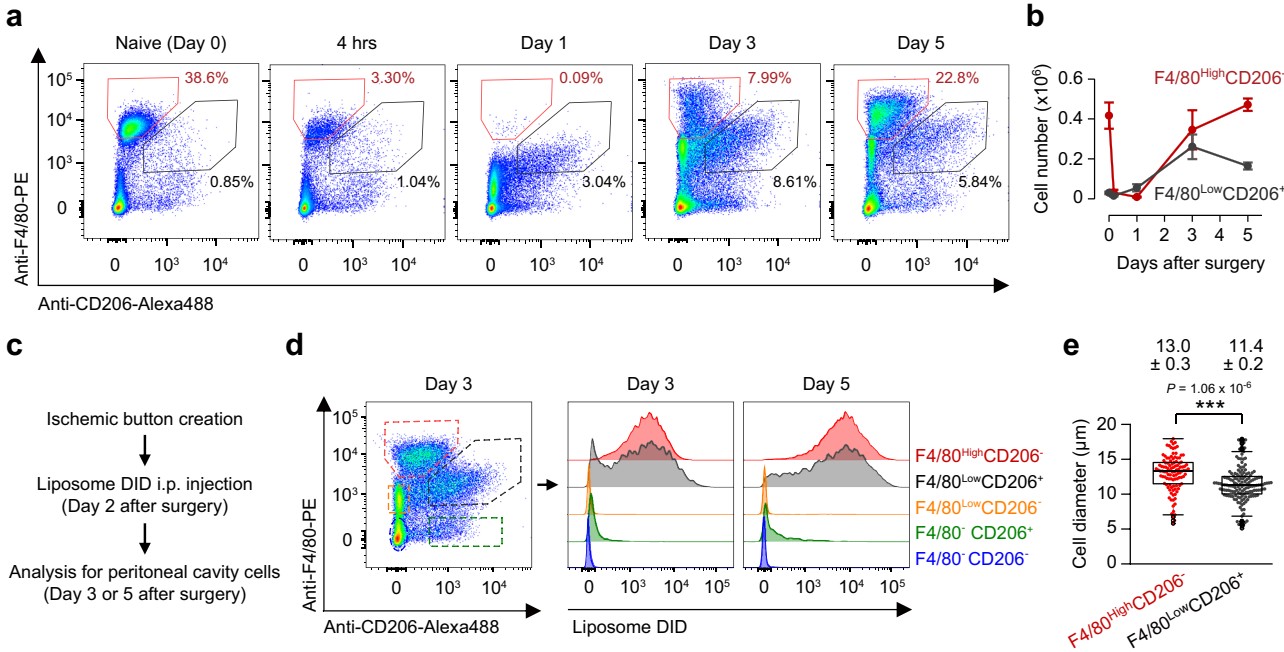

**Fig. 3 Two subsets of peritoneal macrophages exhibit different cellular properties and distinct dynamics post-surgery. a, b** Representative flow cytometry plots (**a**) and cell counts (**b**) of F4/80$^{High}$CD206$^-$ macrophages (red) and F4/80$^{Low}$CD206$^+$ (gray) macrophages in the peritoneal fluid after ischemic button creation in mice. $n = 7$ (Naive; Day 0), 6 (4 h), 8 (Day 1), 8 (Day 3) and 9 (Day 5) mice. See Supplementary Fig. 5 for the gating strategy. Data are shown as the mean ± SEM. **c** Schematic of the phagocytosis uptake assay. **d** Representative flow cytometry histogram showing the phagocytosis activity of peritoneal F4/80$^{High}$CD206$^-$ macrophages (red) and F4/80$^{Low}$CD206$^+$ (gray) macrophages in comparison with F4/80$^{Low}$CD206$^-$ (orange), F4/80$^-$CD206$^+$ (green) and F4/80$^-$CD206$^-$ (blue) cells in the peritoneal cavity. $n = 3$ independent experiments. **e** Diameter of sorted peritoneal F4/80$^{High}$CD206$^-$ and F4/80$^{Low}$CD206$^+$ macrophages. Data are presented as box-and-whisker plots (interquartile ranges (IQRs) as boxes, with the median as a black line and the whiskers extending up to the most extreme points within 1.5-fold IQR). $n = 109$ (F4/80$^{High}$CD206$^-$) cells and 154 (F4/80$^{Low}$CD206$^+$) cells. ***$P < 0.001$, two-tailed Student's $t$-test.

macrophages accumulated on these lesions and appeared to substitute for the disrupted mesothelial barrier albeit in part (50–65% of the surface area covered; Fig. 2d). These macrophages were localized particularly on the surface of exposed fibrin clots, consequently shielding ~60% of the surface area of exposed fibrin clots on day 3 (Supplementary Fig. 2c) and showed a phagocytosis ability (Supplementary Fig. 3a, b). This macrophage barrier was temporal and cleared by day 5 when the mesothelial barrier was reconstituted.

We found that F4/80$^+$ macrophages accumulating on the button surface were negative for CD206 (Supplementary Fig. 4). By contrast, CD206$^+$ cells, which were weakly positive for F4/80, were observed within the ischemic button tissue after day 3, indicating a distinction between these two macrophage subsets. Furthermore, at the lesion where an abdominal adhesion had been formed between the ischemic button and surrounding tissue, reduced accumulation of F4/80$^+$ macrophages on the fibrin clots was noted (Fig. 2e). These results collectively suggest that the cell barrier composed of F4/80$^+$CD206$^-$ macrophages on the exposed fibrin clots may have the potential to contribute to attenuation of post-operative adhesion formation.

**Peritoneal macrophage subsets have distinct dynamics**. We then characterized the surface phenotype of peritoneal macrophages during the course of post-operative abdominal adhesion formation by using flow cytometry. Consistent with the above histological findings, two major macrophage subpopulations, F4/80$^{High}$CD206$^-$ and F4/80$^{Low}$CD206$^+$ subsets, were identified (Fig. 3a, b, and Supplementary Fig. 5). Both of these macrophage subsets, but not F4/80$^{Low}$CD206$^-$, F4/80$^-$CD206$^+$ or F4/80$^-$CD206$^-$ peritoneal cells, showed an evident phagocytosis

activity (Fig. 3c, d). Of note, while F4/80$^{High}$CD206$^-$ macrophages were dominant (~40% of the total peritoneal fluid cells) in the peritoneal fluid in the normal state, they were almost cleared to occupy only 1% of peritoneal fluid cells by 4 h post-surgery (Fig. 3a, b). This clearance was more evident at 24 h post-surgery, at which these cells were hardly detectable. We speculated that this loss of F4/80$^{High}$CD206$^-$ macrophages from the fluid was due to the firm attachment of these cells to fibrin clots on the ischemic button as described in the histological investigation (Fig. 2a–d). The F4/80$^{High}$CD206$^-$ macrophage subset was mostly reconstituted in the peritoneal fluid by day 5. On the other hand, F4/80$^{Low}$CD206$^+$ macrophages were rarely found in the naive peritoneal fluid, but became detectable, occupying 3% of peritoneal fluid cells, on day 1 with a peak at day 3. Throughout the time course studied, the number of F4/80$^{Low}$CD206$^+$ macrophages was similar to or smaller than that of F4/80$^{High}$CD206$^-$ macrophages (Fig. 3b). Both of these peritoneal macrophage subsets exhibited high expressions of tissue repair-related genes and low expressions of pro-inflammatory or angiogenesis-related genes (Supplementary Fig. 6a, b). As compared to the other subset, F4/80$^{High}$CD206$^-$ macrophages were larger in size and exhibited higher mRNA or protein expression of *Gata6, Tgfb2, Ucp1,* and ICAM2 (Fig. 3e and Supplementary Fig. 7a, b), which are reported markers for resident peritoneal macrophages[30–32]. By contrast, the smaller F4/80$^{Low}$CD206$^+$ macrophages exhibited higher CCR2 and MHCII expression (Supplementary Fig. 7b), suggesting their identity as monocyte-derived, recruited macrophages[32,33].

To examine the possibility that the other recruited monocytes/macrophage subsets or granulocytes have the ability to cover the fibrin clot, we have further characterized the peritoneal cavity

cells after ischemic button creation. The results showed that F4/80$^{Low}$CD206$^-$ cells included Ly6C$^{High}$ classical monocytes/macrophages, Ly6C$^{Low}$ non-classical monocytes/macrophages, and Siglec-F$^+$ eosinophils, while Ly6G$^+$ neutrophils were found in F4/80$^-$CD206$^-$ population (Supplementary Fig. 8a, b). However, these cell types did not accumulate on the button surface (Supplementary Fig. 9a). Furthermore, accumulated F4/80$^+$ macrophages on fibrin clots were positive for a resident macrophage marker, ICAM2 (Supplementary Fig. 9b, c). Nonetheless, there remained a concern that recruited monocytes/macrophages or granulocytes would not cover the fibrin clot because of their too-small occurrence at the time of initial adhesion formation (<24 h post-surgery). Therefore, we further assessed the ability of recruited monocytes/macrophages to participate in the cell barrier formation in the "repeated ischemic button creation model" (Supplementary Fig. 10a, b). In this model, we created an additional ischemic button on the right-side peritoneal wall on day 3 after the first ischemic button creation on the left-side peritoneal wall. This model enabled us to investigate the capability of recruited monocytes/macrophages more precisely because the occurrence of these recruited cells in the peritoneal cavity is much more abundant at day 3 compared to the time period for the ordinary cell barrier formation. The obtained result demonstrated that, even with such an increased occurrence, recruited monocytes/macrophages or granulocytes did not contribute to the cell barrier formation. These results together confirm that only resident macrophages have the ability to cover the fibrin clots.

**Depletion of resident macrophages exacerbates adhesions**. We next investigated the functional role of resident peritoneal macrophages in the post-operative adhesion formation by using a depletion strategy. Based on the previous reports[34,35], we adjusted the protocol of pre-injection of clodronate liposomes so that resident F4/80$^{High}$CD206$^-$ peritoneal macrophages were depleted at the time of surgery (Fig. 4a). This protocol did not reduce the increase of recruited F4/80$^{Low}$CD206$^+$ peritoneal macrophages post-surgery (Supplementary Fig. 11a, b). As a result of this clodronate liposome treatment, we found markedly exacerbated abdominal adhesions after ischemic button creation with a reduced cover rate of fibrin clots by F4/80$^+$ macrophages (Fig. 4b–f). Of importance, the degree of the mesothelial disruption or the amount of fibrin deposition was unaffected (Fig. 4g–i and Supplementary Fig. 11c), suggesting that the depletion of resident F4/80$^{High}$CD206$^-$ peritoneal macrophages did not affect fibrin formation or degradation. Consistent with this, the main pro- or anti-fibrinolytic factor, tissue plasminogen activator (tPA)[36] or plasminogen activator inhibitor-1 (PAI-1)[37], in the peritoneal fluid was not affected by the clodronate liposome treatment (Fig. 4j, k). These data indicate that resident F4/80$^{High}$CD206$^-$ peritoneal macrophages contribute to the attenuation of post-operative abdominal adhesion formation through reducing the exposure of fibrin clots as a cell barrier, but not through affecting fibrin production or fibrinolysis.

**Resident macrophages bind fibrin clots via CD11b**. To understand the mechanism by which the fibrin clot attracted resident peritoneal macrophages, we implanted an ex vivo produced fibrin clot using a TISEEL fibrin glue kit (Baxter) into the mouse peritoneal cavity (Supplementary Fig. 12a). As a result, it was observed that a large number of resident peritoneal macrophages accumulated onto the surface of the implanted exogenous fibrin clot, similar to the finding of the fibrin clot on the ischemic button (Fig. 5a, b). A more than half of the exogenous fibrin clot surface was covered by the accumulated resident peritoneal

macrophages. This suggests that the fibrin clot itself has the ability to attract resident peritoneal macrophages.

We hypothesized that CD11b (integrin alpha M), which is known to regulate the binding of microglia and fibrin[38], might play a role in this interaction between resident peritoneal macrophages and fibrin clots. A high-level CD11b expression of resident F4/80$^{High}$CD206$^-$ peritoneal macrophages was confirmed (Supplementary Fig. 12b). Furthermore, proximity ligation assay (PLA) detected clear PLA signals between fibrin and CD11b on F4/80$^+$ macrophages (Fig. 5c), but not between fibrin and F4/80 (Supplementary Fig. 12c). Furthermore, intraperitoneal injection of a blocking antibody to CD11b (clone; 5C6)[39] intensified the abdominal adhesion, which corresponded to the abolished accumulation of F4/80$^+$ macrophages on the fibrin clot (Fig. 5d–g). In addition, the above-mentioned transient disappearance of resident F4/80$^{High}$CD206$^-$ peritoneal macrophages from the peritoneal fluid 1 day after surgery (Fig. 3a, b) was inhibited by blocking CD11b, while the occurrence of recruited F4/80$^{Low}$CD206$^+$ peritoneal macrophages in the peritoneal fluid was unaffected (Supplementary Fig. 12d, e). Thus, we conclude that accumulation of resident F4/80$^{High}$CD206$^-$ peritoneal macrophages on the fibrin clot exposed on the damaged peritoneum was achieved directly via CD11b.

**IL-4 reduces adhesions by enhancing macrophage barriers**. Having identified the cell barrier function of resident F4/80$^{High}$CD206$^-$ peritoneal macrophages, this naturally-occurring process was not sufficient in preventing abdominal adhesions (Fig. 1 and Supplementary Fig. 1b, c). We thus explored if forced augmentation of resident F4/80$^{High}$CD206$^-$ peritoneal macrophages would result in robust prevention of post-operative adhesion formation. We confirmed that intraperitoneal administration of long-acting IL-4 complex (IL-4c)[40] increased the number of resident F4/80$^{High}$CD206$^-$ peritoneal macrophages approximately three times (Fig. 6a, b). This IL-4c treatment indeed resulted in a markedly attenuated formation of post-operative adhesions with increased coverage of the exposed fibrin clots by F4/80$^+$ macrophages (Fig. 6c–f). Histological assessments demonstrated that mesothelial cells on the peritoneal membrane or fibrin deposition on the ischemic button lesion were unaffected by IL-4c injection (Fig. 6g–i). In addition, IL-4c administration had no effect on other types of immune cells in the peritoneal cavity or within the ischemic button tissue (Supplementary Fig. 13a–c). Supporting this observation, IL-4c administration did not affect the Th1/Th2-related cytokine production levels in the peritoneal fluid 3 days after surgery, while a macrophage-derived chemokine, CCL6/C10[41], was up-regulated (Supplementary Fig. 14a, b).

Consistent with the previous report that IL-4 promotes not only resident macrophage proliferation but also recruited macrophage polarization[32], IL-4c administration also increased the number of recruited F4/80$^{Low}$CD206$^+$ macrophages in the peritoneal fluid (Fig. 6a, b), while this cell number was less than one-third of the F4/80$^{High}$CD206$^-$ macrophage number. To exclude the possibility that recruited F4/80$^{Low}$CD206$^+$ macrophages contributed to the attenuation of post-operative adhesions, we isolated each macrophage subsets from IL-4c injected mice and performed adoptive transfer into the peritoneal cavity of a syngeneic recipient mouse at the time of the ischemic button creation. As we expected, adoptive transfer of resident F4/80$^{High}$CD206$^-$ peritoneal macrophages, but not recruited F4/80$^{Low}$CD206$^+$ peritoneal macrophages, significantly reduced post-operative adhesions (Fig. 6j, k). Accumulation of a sizeable number of transferred resident F4/80$^{High}$CD206$^-$ peritoneal macrophages on the surface of the ischemic button was

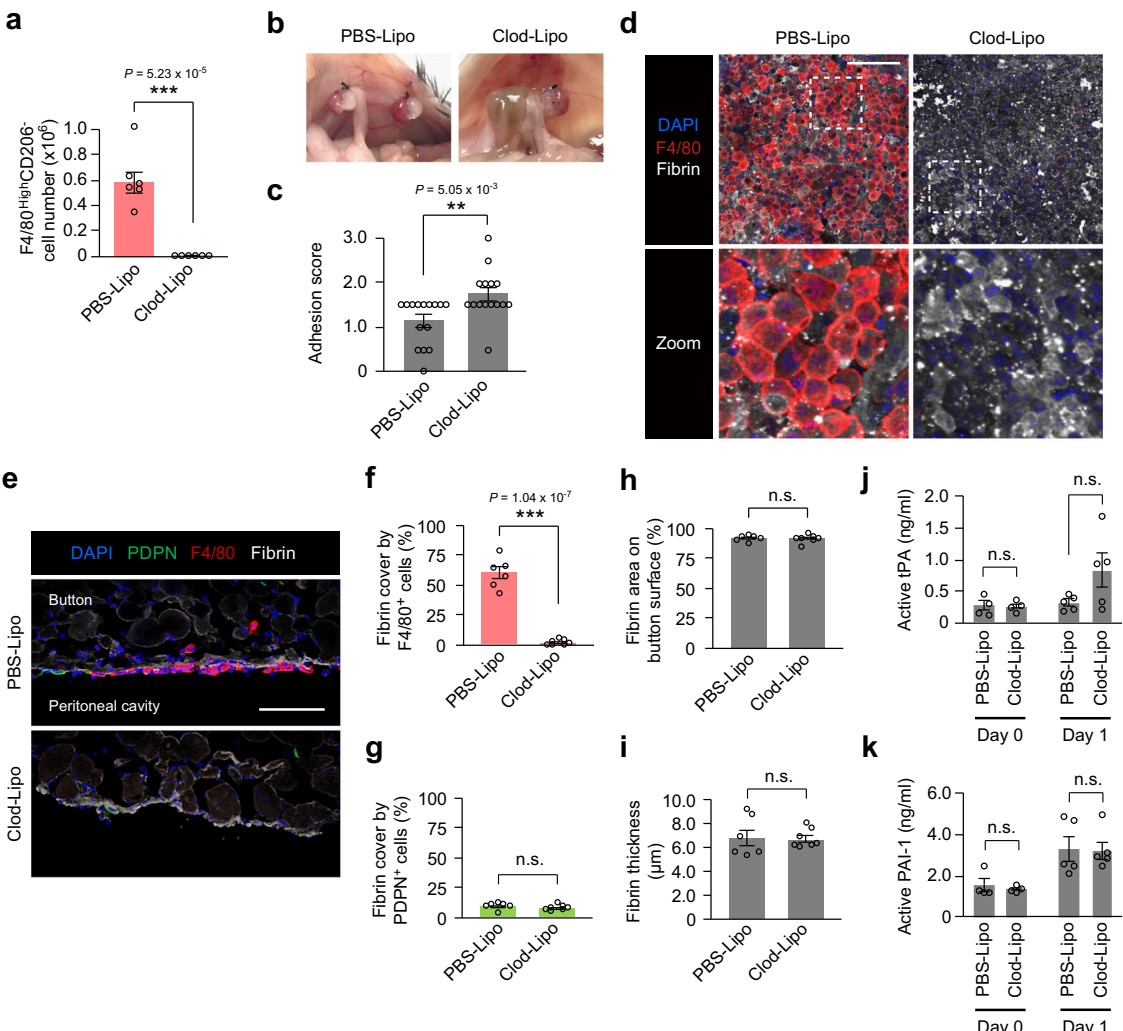

**Fig. 4 Depletion of resident F4/80^HighCD206⁻ peritoneal macrophages exacerbates post-operative abdominal adhesion formation. a** The cell number of resident F4/80^HighCD206⁻ peritoneal macrophages in the peritoneal cavity on day 2 after intraperitoneal injection of clodronate liposomes compared to the control PBS-liposome injection. $n = 6$ mice in each group. **b, c** The visual appearance of the abdominal adhesion formation (**b**) and the adhesion score (**c**) on day 1 post-ischemic button creation with an intraperitoneal injection of clodronate liposomes or PBS-liposomes two days before surgery. $n = 15$ mice in each group. **d** Representative images of the whole-mount immunohistostaining of the ischemic button on day 1 after surgery with pre-injection of clodronate liposomes or PBS-liposomes. $n = 3$ mice in each group. Scale bars, 100 μm. **e–i** Representative cross-section immunofluorescent images and quantification of the ischemic button on day 1 after surgery with the injection of clodronate liposomes or PBS-liposomes. The bar graphs show the coverage of exposed fibrin clots by F4/80⁺ cells (**f**) or PDPN⁺ cells (**g**). The fibrin area (**h**) or thickness (**i**) on the ischemic button surface were also histologically quantified. $n = 6$ (PBS-Lipo group) and 7 (Clod-Lipo group) mice. Scale bars, 100 μm. **j, k** ELISA measurements for active tPA protein (**j**) or active PAI-1 protein (**k**) in the peritoneal fluid after surgery with pre-injection of clodronate liposomes or PBS-liposomes. $n = 4$ (Day 0) and 5 (Day 1) mice in each group. Data represent the mean ± SEM. **$P < 0.01$, ***$P < 0.001$, ns not significant, two-tailed student's $t$-test.

confirmed (Fig. 6l). These data proposed that the IL-4-based treatment would achieve substantial prevention of post-operative adhesion formation through augmentation of the resident peritoneal macrophage barrier that shields exposed fibrin clots.

## Discussion

Despite its enormous clinical importance, cellular and molecular processes underlying the formation and/or prevention of post-operative adhesions remains poorly understood[17]. Here, in a mouse model, we reveal a previously unidentified role of resident peritoneal macrophages to attenuate post-operative adhesion formation through the generation of a cell barrier (Fig. 7). Shortly after surgical insult, sticky fibrin clots are formed on the damaged peritoneal membrane, where the mesothelial barrier is lost. These fibrin clots capture resident F4/80^HighCD206⁻ peritoneal

macrophages on their surface through CD11b-mediated binding. This macrophage accumulation results in the formation of a cell barrier that acts to shield the adhesive fibrin clot from the surrounding tissue in place of the lost mesothelial barrier, potentially attenuating the adhesion formation. The macrophage barrier is formed promptly by day 1 post-surgery and is temporal being replaced by the reconstructed mesothelial cell barrier by day 5. Compared to the complete mesothelial barrier, the macrophage barrier is insufficient; it covers only ~60% of the exposed fibrin clot area. As a result, post-operative adhesions are developed at the lesions that have an inadequate macrophage barrier. Of clinical impact, the resident macrophage barrier can be augmented by IL-4-based treatment or adoptive cell transfer, which leads to robust prevention of the adhesion formation, proposing an innovative anti-adhesion strategy. In addition, recruiting peritoneal macrophages are not involved in the

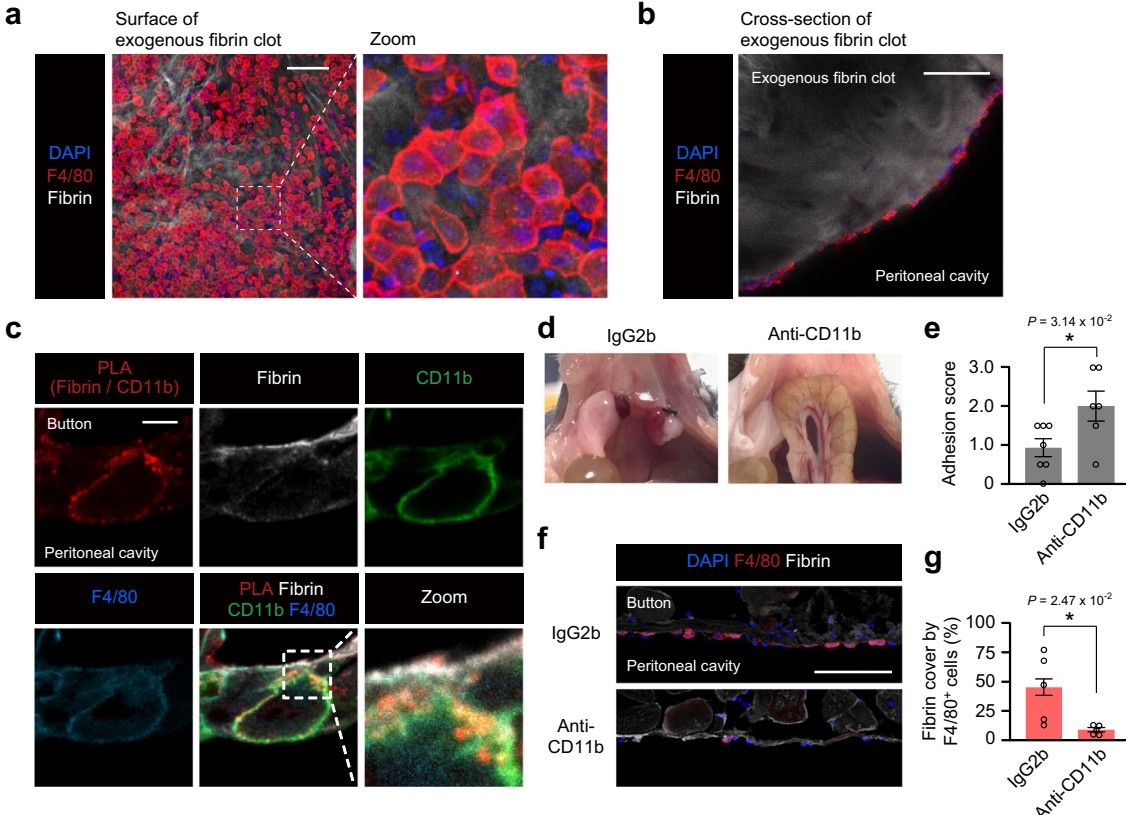

**Fig. 5 Resident F4/80^HighCD206^− peritoneal macrophages bind fibrin clots via CD11b. a, b** Representative confocal image of the whole-mount (**a**) and cross-section (**b**) immunofluorescent staining of the implanted exogenous fibrin clot on day 1 after implantation. $n = 3$ mice. Scale bars, 100 µm. (**c**) Representative confocal images of PLA between CD11b of F4/80^+ macrophages and fibrin on the ischemic button surface. PLA signals are shown in red. $n = 3$ independent experiments. Scale bars, 5 µm. **d, e** Representative images of abdominal adhesions (**d**) and adhesion scores (**e**) on day 1 post-ischemic button creation with an intraperitoneal injection of control IgG2b antibody (IgG2b group, $n = 6$ mice) or anti-CD11b blocking antibody (Anti-CD11b group, $n = 7$ mice). **f, g** Representative cross-section immunohistostaining images of the ischemic button (**f**) and quantification of the coverage of exposed fibrin clots by F4/80^+ cells (**g**) on day 1 after surgery with an intraperitoneal injection of control IgG2b antibody or anti-CD11b blocking antibody. $n = 5$ mice in each group. Scale bar, 100 µm. Data represent the mean ± SEM. *$P < 0.05$, two-tailed Student's $t$-test.

protective cell barrier mechanism, shedding light on a previously unknown biological insight regarding the complex diversity of macrophages.

We here demonstrate that the accumulation of resident peritoneal macrophages on the injured peritoneum is mediated through binding between CD11b and fibrin. There is a report describing that peritoneal macrophages are attracted to the damaged tissue through sensing extracellular ATP and contribute to the removal of dead cells in a heat-induced liver injury model in mice[34]. However, we observe that implantation of an externally-produced fibrin clot, which is unlikely to release a significant level of ATP, results in macrophage accumulation on its surface, similar to that observed in the ischemic button model. Moreover, in addition to the positive PLA signals between CD11b and fibrin, the inhibition of CD11b-fibrin interaction using a blocking antibody abolishes the accumulation of resident macrophages onto the damaged peritoneum of the ischemic button. Taken together, although a possible role of extracellular ATP and other factors cannot be fully excluded, we conclude that the CD11b-fibrin binding is a major mechanism by which resident peritoneal macrophages accumulated onto the injured peritoneum in our model of post-operative adhesion.

One of the interesting findings in this study is that resident peritoneal macrophages drastically but transiently disappeared from the peritoneal fluid after adhesion-inducing surgery. A similar phenomenon has been reported as the macrophage disappearance reaction (MDR) in the setting of intra-abdominal infection[42,43]. It is also known that intraperitoneal injection of lipopolysaccharide[30], zymosan[35], or thioglycolate[44] induces the MDR and that the MDR is suppressed by pretreatment with anticoagulants, including warfarin and heparin[35,42]. The underpinning mechanism of the MDR is thought to be macrophage accumulation to a foreign body or migration into the lymphatic vessels, rather than the death of macrophages[35,43]. Our study reports that the MDR occurs during post-operative adhesion formation as well. Resident F4/80^HighCD206^− peritoneal macrophages that occupy around 40% of the total peritoneal fluid cells in the normal status are almost cleared as early as day 1 after surgery. The series of histological examinations in this study suggest that this clearance from the peritoneal fluid is due to the attachment of these cells to the fibrin clots on the ischemic button. By day 5 after surgery, the population of resident F4/80^HighCD206^− peritoneal macrophages is reconstituted in the peritoneal fluid. Although the source of these reconstituting macrophages remains uncertain, it could be their detachment from the fibrin clots, migration from the peritoneal organs[45], differentiation from recruited monocytes, and/or phenotype change from recruited F4/80^LowCD206^+ peritoneal macrophages or other cells types[46–48].

Peritoneal macrophages are classified into two major distinct subsets, the large peritoneal macrophage (LPM) and small peritoneal macrophage (SPM), basically according to their size[33].

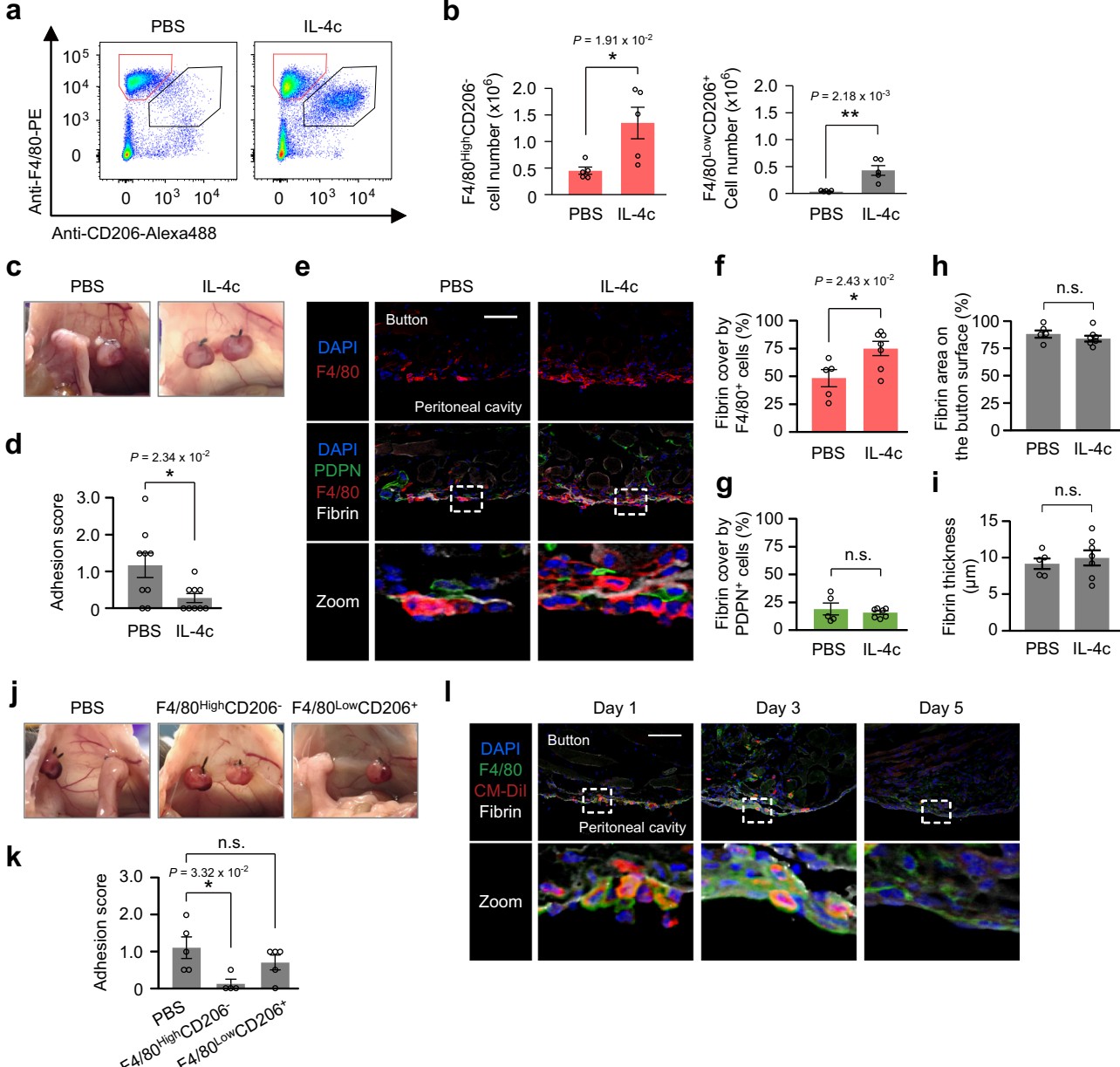

**Fig. 6 IL-4c administration augments the macrophage barrier and attenuates post-operative adhesion formation. a, b** Representative flow cytometry plots (**a**) and cell numbers (**b**) of peritoneal macrophage subsets day 2 after injection of IL-4c or PBS only. $n = 5$ mice in each group. **c, d** The representative appearance of post-operative adhesions (**c**) and the adhesion score (**d**) on day 3 post-surgery. IL-4c or PBS was administered into the peritoneal cavity immediately after ischemic button creation surgery. $n = 9$ mice per group. **e–i** Representative immunofluorescence images and quantification of cross-sections of the ischemic button on day 3 after surgery with IL-4c or PBS administration. The bar graphs show the coverage of exposed fibrin clots by F4/80$^+$ cells (**f**) or PDPN$^+$ cells (**g**). The fibrin area (**h**) or thickness (**i**) on the ischemic button surface were also histologically measured. $n = 5$ (PBS) and 7 (IL-4c) mice. Scale bars, 100 μm. **j, k** Representative images of post-operative abdominal adhesions (**j**) and calculated adhesion scores (**k**) on day 5 after ischemic button creation with intraperitoneal adoptive transfer of syngeneic mouse F4/80$^{High}$CD206$^-$ or F4/80$^{Low}$CD206$^+$ peritoneal macrophages, or PBS administration. $n = 5$ (PBS), 4 (F4/80$^{High}$CD206$^-$) and 5 (F4/80$^{Low}$CD206$^+$) mice. **l** Representative immunofluorescence images of the ischemic button with adoptive transfer of resident F4/80$^{High}$CD206$^-$ peritoneal macrophages (pre-labeled with CM-DiI; red). $n = 2$ independent experiments. Scale bar, 100 μm. Data represent the mean ± SEM. *$P < 0.05$, **$P < 0.01$, ns not significant, one-way ANOVA and Tukey's post hoc test in (**k**) or two-tailed student's $t$-test in others.

LPMs are the resident macrophages, which are dominant at the steady-state and maintained in the peritoneal cavity through self-renewal, and have an F4/80$^{High}$, CD11b$^{High}$, MHCII$^{Low}$, GATA6$^{High}$, and CD206$^-$ phenotype[46,49]. By contrast, SPMs are the recruited macrophages, which are derived from circulating monocytes and increase in number in response to an inflammatory stimulus in a CCR2 dependent manner[30]. These cells exhibit an F4/80$^{Low}$, CD11b$^{Low}$, MHCII$^{High}$, GATA6$^{Low}$, and CD206$^+$ phenotype. Sexually dimorphic replenishment of LPM by bone marrow-derived cells, which is high in males and very low in females, has also been reported[48]. In addition, there is information available regarding the functional diversity of these subsets. For instance, LPMs regulate the peritoneal B-1 cell-mediated gut IgA production through TGFβ2 under the normal

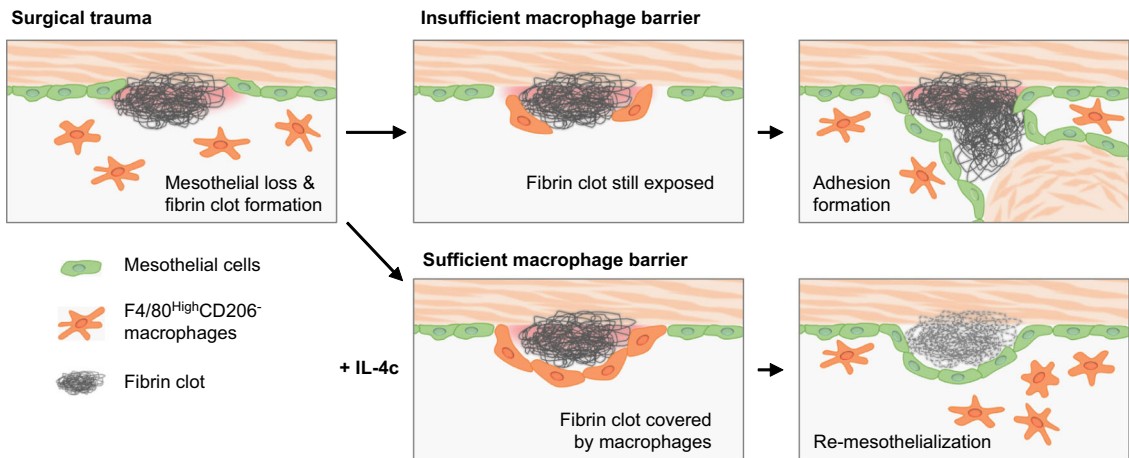

**Fig. 7 The function of macrophage barriers during adhesion formation.** The schematic illustrates the role of resident macrophage barriers during post-operative abdominal adhesion formation. Upon injury of the peritoneal membrane, resident F4/80[High]CD206[−] peritoneal macrophages promptly accumulate onto the lesion where fibrin clots are exposed. These accumulated macrophages form an anti-adhesion cell barrier to shield the sticky fibrin clots from neighboring tissues. This macrophage barrier is, however, usually insufficient, allowing the formation of adhesions (Upper panels). However, when this cell barrier is sufficiently strengthened i.e., using IL-4c treatment, post-operative adhesions are robustly prevented (Lower panels).

condition[30]. SPMs have an ability to present antigens to naive CD4[+] T cells[50], suggesting that SPMs induce local T-cell priming during peritoneal infection or inflammation. In our study, F4/80[High]CD206[−] and F4/80[Low]CD206[+] macrophage subsets exhibited comparable characters to LPMs and SPMs, respectively, in terms of morphology and gene/protein expression[30,33,51]. However, further investigation is needed to define these macrophages as LPMs and SPMs.

We observed that sizeable numbers of CD11b[+]CCR2[+]Ly6C[High] classical monocytes/macrophages and CD11b[+]CCR2[+]Ly6C[Low] non-classical monocytes/macrophages, which were F4/80[Low]CD206[−], appeared in the peritoneal cavity at day 3 post-surgery. Unlike F4/80[High]CD206[−] and F4/80[Low]CD206[+] macrophage subsets, these F4/80[Low]CD206[−] cell populations did not present a phagocytic ability, suggesting that they were unlikely to be mature macrophages. Furthermore, our results showed that CCR2[+] recruited monocytes/macrophages did not have the ability to form an anti-adhesion cell barrier. On the other hand, there is a report showing that pre-treatment of rabbits with proteose peptone or transplantation of proteose peptone-elicited peritoneal macrophages protect against adhesion formation, suggesting that monocyte-derived inflammatory macrophages might offer an anti-adhesion effect in a certain condition[21]. It is speculated that this effect of monocyte-derived macrophages may be attained through a different mechanism(s) from cell barrier formation, including modulation of inflammation, fibrin formation, fibrinolysis and/or vacuolization, and fibrosis of adhesion tissues[21–25].

IL-4 is known to play pleiotropic roles in the immune system[52]. We found that intraperitoneal IL-4c administration attenuated post-operative adhesion formation by increasing the number of F4/80[High]CD206[−] resident peritoneal macrophages. Adoptive transfer of IL-4c-treated F4/80[High]CD206[−] macrophages resulted in comparable attenuation of adhesions. On the other hand, IL-4c administration did not affect the number of non-macrophage cell types including lymphocytes or the major Th1/Th2-related cytokine levels in the peritoneal fluid. Therefore, we consider that the resident peritoneal macrophage barrier is the central mechanism for the anti-adhesion effect of IL-4c administration. However, we cannot exclude a possibility of functional alterations of these cells in response to IL-4, which would modulate a process of adhesion formation. Further detailed cell-type dependent

responses to IL-4 and the resulting effects on adhesion formation could be elucidated by using cell-type-specific IL-4 receptor knockout mice. Of note, a previous study has reported that IL-4 knockout did not affect post-operative adhesions in mice[24], indicating that endogenous IL-4 is unlikely to play a critical role in the adhesion formation.

The findings obtained in this study have significant clinical implications in developing an innovative anti-adhesion strategy. To prevent post-operative adhesion formation, a range of pharmacological agents, including fibrinolytic agents, non-steroidal anti-inflammatory drugs, steroids, antioxidants, histamine antagonists, and statins, were tested; however, none of them has been clinically successful to date[53]. Instead, implantation/administration of a biomaterial barrier product is currently conducted for the prevention of adhesions[7]. Nonetheless, this method has obvious limitations in its efficacy and practicability[8–10]. Macrophage barriers are likely to have multiple important advantages over the biomaterial-based barrier. The formation of a macrophage barrier relies on a natural, physiological healing process, while an artificial biomaterial barrier is associated with possible foreign body reactions[54]. The macrophage barrier has no risk of physical displacement, unlike a biomaterial barrier[9,10]. In addition, the mobility and flexibility of macrophages enable the formation of a cell barrier at any location throughout the peritoneal cavity, including difficult-to-access localizations, i.e., deep inside of the pelvic cavity or between intestines. Furthermore, sufficient macrophage barriers can be achieved simply by injection of a pharmacological reagent, i.e., IL-4c, and is thereby accessible in any type of surgical procedures, including open, laparoscopic, or robotic surgeries. Further development of this innovative treatment to prevent post-operative adhesions, which is based on the previously unknown concept of the macrophage barrier, is warranted. To this end, it is essential to confirm the universality of the results obtained in this study. We here used the mouse ischemic button model because it has an important advantage in inducing reproducible intra-abdominal adhesions compared to other models as previously demonstrated[55]. Indeed, this model offered reliable and adequate adhesion formation to our study. As the next step toward clinical application, investigations using a different model, particularly a large animal model, and also research with human cells will be required.

## Methods

**Study approval**. All investigations using living animals conformed to the Principles of Laboratory Animal Care formulated by the National Society for Medical Research and to NIH guidelines (Guide for the Care and Use of Laboratory Animals. National Academies Press 1996) and performed with the approval of the ethics committee of the Queen Mary University of London and the UK Home Office (Project License PPL70/8503). All in vivo procedures were performed by UK Home Office Personal License holders. In vivo and in vitro procedures and assessments were blinded wherever possible.

**Post-operative abdominal adhesion model**. Ten- to twelve-week-old male C57BL/6 mice were purchased from Charles River Laboratories UK and used in the experiments. These mice were maintained in a specific pathogen-free room under a 12-h light/12-dark cycle, 20–24 °C temperature, and 45–55% humidity with free access to food and water in our animal facility. Surgery was performed under aseptic conditions, and all efforts were made to minimize animal suffering. Anesthesia was induced by 2.0% isoflurane inhalation. Abdominal adhesion formation was induced by the construction of two ischemic buttons on the peritoneal wall, as previously described[27]. After a small median laparotomy, two ischemic buttons, ~5 mm in diameter each and spaced ~10 mm apart, were created on the right-side peritoneal wall by clamping and ligating with a 5–0 Mersilk suture (W2500T, Ethicon). The abdomen was closed with a double-layered suture of the peritoneum and skin with a 5–0 vicryl absorbable suture (W9915, Ethicon). They were allowed to recover on a heating pad and monitored daily until euthanasia. At the chosen time point, the mice were sacrificed with carbon dioxide ($CO_2$) inhalation to evaluate abdominal adhesion formation, and peritoneal fluid/tissue samples collected. According to the adhesion scoring system[4,28], the adhesion scores for each ischemic button (Lower; tail side, Upper; head side) were determined as follows: 0, no adhesion; 1, filmy adhesion (separated easily by blunt dissection); 2, firm adhesion (separated by aggressive blunt dissection); 3, dense adhesion (separable only by sharp dissection). Adhesion scores were validated in a blinded manner, and the average of lower and upper ischemic button scores was calculated (Supplementary Fig. 1a). For the repeated ischemic button creation model (Supplementary Fig. 10a), two ischemic buttons were first created on the left-side peritoneal wall of a mouse as described above. Three days after this first surgery, the abdomen was re-opened, and an additional ischemic button was created on the right-side peritoneal wall.

**Hematoxylin and eosin staining and immunohistolabeling**. After harvesting, tissues were fixed in 4% paraformaldehyde (PFA) overnight at 4 °C and were embedded and frozen in O.C.T. compound (VWR International).

For H&E staining of ischemic buttons and adhesion tissues, 6 μm thick frozen sections were cut and incubated in Mayer's hematoxylin solution (Sigma-Aldrich, MHS16) for 15 min and Eosin Y solution (Sigma-Aldrich, HT110116) for 1 min. After dehydration through increasing concentrations of ethanol to xylene, the sections were mounted using the DPX mounting medium (VWR International, 13512). The digital images were acquired with an All-in-One microscope (BZ-8000; KEYENCE) using a 2× or 20× objective lens (1360 × 1024 pixels).

For immunohistolabeling, frozen sections were cut at 6 μm thick and pre-blocked with the blocking buffer (PBS plus 5% BSA and 0.1% Tween 20) for 30 min at room temperature. To block endogenous biotin, if required, samples were pre-blocked with streptavidin buffer (PBS plus 0.1 mg/ml Streptavidin and 0.1% Tween 20) for 30 min and biotin buffer (PBS plus 0.5 mg/ml Biotin and 0.1% Tween-20) for 30 min at room temperature. Samples were labeled with the primary antibodies overnight at 4 °C. Information about primary antibodies and dilutions used in the assay can be found in "Supplementary Table 1". The anti-laminin antibody (1:1000 dilution, Sigma, L9393) was conjugated with Alexa Fluor 488 fluorophores using labeling kits (Life Technologies, A20181) according to the manufacturer's instruction. After rinsing in PBS three times for 5 min, the sections were next incubated with 4′,6-diamidino-2-phenylindole (DAPI) (Sigma-Aldrich, D9542) and the appropriate fluorophore-conjugated secondary antibodies or AlexaFluor 546-conjugated Streptavidin (1:300 dilution, Invitrogen, S11225) for 1 h at the room temperature. Stained sections were mounted with DAKO Fluorescence Mounting Medium (Agilent, S302380-2), and the digital images were acquired, and pseudo-colored with an All-in-One microscope (BZ-8000; KEYENCE) using a 10× or 20× objective lens (1360 × 1024 pixels).

All image analysis was performed by importing images as TIFF files into ImageJ version 1.50i software (NIH). To visualize a whole image of the ischemic button (Fig. 2a), sixteen images were reconstructed by using the MosaicJ plugin. For the quantification of the cover ratio by F4/80+ macrophages or PDPN+ mesothelial cells on the ischemic button and fibrin clots, F4/80 or PDPN positive area was quantified as the percentage of the surface of peritoneal area or fibrin positive area. For quantification of fibrin formation, the fibrin area and thickness were quantified as the percentage or average thickness of the fibrin positive area on the ischemic button (Supplementary Fig. 11c). Images containing the surface area of the ischemic button, excluded adhesion area, were acquired at three independent regions and analyzed. A color threshold function of ImageJ was applied to measure macrophages, mesothelial cells, and fibrin-clot area.

**Peritoneal cell isolation and flow cytometry analysis**. After mice were sacrificed using $CO_2$ inhalation, peritoneal cells were obtained by washing the peritoneal cavity with ice-cold PBS four times. Erythrocytes were removed by incubating with red blood cell lysis buffer (Biolegend, 420301) at 5 min on ice. Total live cell number was counted using a Countess II Automated Cell Counter (Invitrogen, AMQAX1000) with Trypan Blue staining. After centrifuging, peritoneal cells were resuspended in FACS buffer (HBSS plus 2 mM EDTA and 0.5% BSA) and pre-incubated with an anti-mouse CD16/CD32 antibody (clone: 93) (1:100 dilution, Invitrogen, 14–0161–85) to block the Fc receptor. Dead cells and debris were excluded by forwarding scatter/side scatter (FSC/SSC) and DAPI staining (1:500 dilution). Phenotype analysis was performed by staining peritoneal cells with the fluorophore-conjugated antibodies for 3 h at 4 °C. Information about primary antibodies and dilutions used in the assay can be found in "Supplementary Table 1". In order to accurately identify the positive signal, appropriate isotype antibodies were used as a negative control. Flow cytometric analyses were performed using a BD LSRFortessa cell analyzer (BD Biosciences) with FlowJo software version 10 (Tree Star). Exemplifying the gating strategy is provided in the Supplementary Information (Supplementary Fig. 5). Cell sorting was performed using a FACS Aria II (BD Biosciences). To measure the cell size of individual F4/80$^{High}$CD206$^-$ and F4/80$^{Low}$CD206$^+$ macrophages, the digital images of sorted cells were acquired with an All-in-One microscope (BZ-8000; KEYENCE) using a 20× objective lens (1360 × 1024 pixels). Cell diameter was determined using ImageJ software.

**Whole-mount staining**. Tissues or exogenous fibrin clots were fixed in 4% PFA overnight at 4 °C and permeabilized and blocked in PBS containing 0.5% Triton X-100 and 5% BSA for 4 h at 4 °C. Subsequently, samples were labeled with primary antibodies overnight at 4 °C. After rinsing in PBS three times for 5 min, the sections were next incubated with DAPI and the appropriate fluorophore-conjugated secondary antibodies overnight at 4 °C. Confocal images were captured in z-stacks of 9–13 planes and reconstructed with the maximum intensity projection using ZEN black 3.0 lite software (Carl Zeiss). All antibodies using whole-mount staining are described in Supplementary Table 1.

**Administration of liposomes, blocking antibodies and IL-4c**. For macrophage depletion, 50 μg of clodronate liposomes (Clod-Lipo) or control liposomes (PBS-Lipo) (Encapsula NanoSciences, CLD-8901) were injected intraperitoneally 2 days before surgery. In order to determine the phagocytosis ability of peritoneal cavity cells, 50 μg of Fluoroliposome-DiD (Encapsula Nano Sciences, CLD-8913) were injected intraperitoneally 2 days after surgery. For blocking macrophage-fibrin interaction, 50 μg anti-CD11b neutralizing antibody (clone: 5C6) (Invitrogen, MA5-16528) or rat IgG2b control antibody (clone: eB149/10H5; eBioscience,16–4031–85) was injected intraperitoneally 3 h before and immediately after surgery. For increasing macrophages, IL-4c (5 μg recombinant IL-4 (PeproTech, 214–14) and 25 μg stabilizing monoclonal anti-IL-4 antibody (clone: BVD4-1D11; BD Biosciences, 554387)) dissolved in 100 μl PBS were intraperitoneally injected to normal mice or administrated into the peritoneal cavity immediately after the ischemic button creation. An equivalent volume of PBS was injected as a control.

**Active tPA and PAI-1 measurements**. Peritoneal fluid from the adhesion model mice that were pre-injected with clodronate liposomes or control liposomes was collected through washing the peritoneal cavity with 500 μl ice-cold PBS. Protein concentrations of the peritoneal fluid were determined using a DC Protein Assay (Bio-Rad, 5000112). ELISA was performed according to the manufacturer's instructions of mouse active tPA functional assay ELISA kit (Innovative Research, IMSTPAKTA) and active mouse PAI-1 functional assay ELISA kit (Innovative Research, IMSPAI1KTA).

**Exogenous fibrin clot implantation into the mouse abdomen**. Exogenous fibrin clots for implantation were produced using TISSEEL (Baxter, 1502243). Briefly, 10 μl of component 1 (Human fibrinogen, 72–110 mg/ml; Aprotinin (synthetic), 3000 KIU/ml) and 10 μl of component 2 (Human thrombin, 500 IU/ml; Calcium chloride dihydrate, 40 μmol/ml) were mixed on a sliding glass using a pipette tip. For easy identification of exogenous fibrin clots in histological analyses, 10 mg/ml Alexa Fluor 647-conjugated human fibrinogen (Invitrogen, F35200) was mixed in component 1 (1:500 dilution). After fibrin clot formation, the exogenous clots (~5 mm in diameter) were washed five times with PBS to remove excess salt and implanted into the peritoneal cavity of a mouse through a laparotomy (Supplementary Fig. 12a).

**Proximity ligation assay**. Interaction between CD11b and fibrin was examined using reagents from the Duolink proximity ligation assay (Sigma-Aldrich, DUO92106) following the manufacturer's instructions. Briefly, 4% PFA-fixed frozen sections were cut at 6 μm thick and pre-blocked with the blocking buffer (PBS plus 5% BSA and 0.1% Tween 20) for 30 min at room temperature. Samples were labeled with anti-CD11b antibody (clone: M1/70; 1:200 dilution, eBioscience, 14–0112–85) and anti-fibrin antibody (1:500 dilution, Abcam, ab34269) overnight at 4 °C. After rinsing in PBS three times for 5 min, the sections were incubated with Goat AlexaFluor 488-conjugated anti-rat IgG antibody (1:300 dilution, Invitrogen, A11006) for 1 h at room temperature. Slides were washed and incubated with a PLA probe antibody, which is conjugated with PLUS or MINUS short nucleotide

sequences, for 1 h at 37 °C. PLUS and MINUS oligonucleotides were ligated and amplified by rolling circle PCR with fluorescent nucleotides for 90 min at 37 °C. After rolling circle amplification, slides were washed and incubated with APC-anti-F4/80 antibody (clone: BM8; 1:200 dilution, Biolegend, 123116) and Goat Alexa-Fluor 405-conjugated anti-rabbit IgG antibody (1:300 dilution, Invitrogen, A31556) for 1 h at room temperature. F4/80 and fibrin interaction was examined as a negative control, in the same manner, using anti-F4/80 antibody (clone: BM8; 1:200 dilution, eBioscience, 14–4801–81) and APC-anti-CD11b antibody (clone: M1/70; 1:200 dilution, eBioscience, 17–0112–82). PLA was visualized using an inverted Zeiss 800 (Carl Zeiss) confocal laser scanning microscope equipped with 63× (1.4 NA) objective and solid-state laser diodes (405, 488, 561, and 640 nm excitation wavelengths).

**Adopted macrophage transfer.** Peritoneal cavity cells were collected from four IL-4c administrated, syngeneic donor mice on day 5 post-ischemic button creation. F4/80$^{High}$CD206$^-$ and F4/80$^{Low}$CD206$^+$ macrophages were isolated using a FACS Aria II (BD Biosciences), as described above. We transferred 400,000 cells in 100 μl PBS per recipient mouse directly into the peritoneal cavity immediately after surgery. After 5 days, the mice were sacrificed and adhesion scores evaluated. For enabling histological tracing of transferred macrophages, isolated F4/80$^{High}$CD206$^-$ peritoneal macrophages were labeled with CM-DiI before transplantation (Invitrogen, C7000) according to the manufacturer's recommendations.

**RNA extraction and real-time PCR.** Total RNA was extracted using the PicoPure™ RNA Isolation Kit (Applied Biosystems, KIT0204) from F4/80$^{High}$CD206$^-$ and F4/80$^{Low}$CD206$^+$ macrophages, which were isolated from the peritoneal cavity 5 days post-surgery using a cell sorter as described above. The RNA concentration and quality were measured using the DS-11 Spectrophotometer (DeNovix). cDNA was prepared using the High-Capacity cDNA Reverse Transcription Kit (Applied Biosystems, 4368814) according to the manufacturer's instruction. Real-time PCR was performed by Quant studio 7 qPCR machine (Applied Biosystems) with a PowerUP SYBR Green Master Mix (Applied Biosystems, A25742). Bone marrow mononuclear cells (BMMNCs) were collected from the femurs and tibiae of normal mice as previously described[56] and were used as a control. Fold changes were calculated by relative quantification based on the second derivative method. Gene expression levels were normalized by the *Gapdh* gene. All primers are described in "Supplementary Table 2".

**Cytokine profiling.** The peritoneal fluid was collected through washing the peritoneal cavity with 500 μl ice-cold PBS of the mice on day 3 post-ischemic button creation surgery with IL-4c or PBS administration. After determining the protein concentration using a DC Protein Assay (Bio-Rad, 5000112), the peritoneal fluid containing 20 μg total protein was pooled from 6 mice (total 120 μg protein for each group). Cytokine profiling was performed according to the manufacturer's instructions of Mouse XL Cytokine Array Kit (R&D Systems, ARY028). Briefly, following the block with blocking buffer at room temperature for 1 h, the membranes were incubated with pooled peritoneal fluids overnight at 4 °C. After rinsing with washing buffer three times for 10 min, membranes were incubated with a biotinylated detection antibody cocktail at room temperature for 1 h and then incubated with streptavidin–horseradish peroxidase for 30 min. Membranes were exposed to the peroxidase substrate and measured with the Azure c600 Imager (Azure Biosystems). The images were analyzed using ImageJ software.

**Statistics.** All statistical tests were performed using RStudio version 1.0.153 software and GraphPad Prism 8 software (GraphPad Software). Data represent the mean ± SEM. For comparisons between multiple groups, one-way ANOVA was performed, followed by Tukey's post hoc test. Two groups were compared using the two-tailed, unpaired Student's *t*-test.

**Reporting summary.** Further information on research design is available in the Nature Research Reporting Summary linked to this article.

## Data availability

Data that support the findings of this study are available within the article and Supplementary Information or from the corresponding author upon reasonable request. Raw source data are present in the Source Data file Source data are provided with this paper.

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

## Acknowledgements

We thank Professors Sussan Nourshargh and Antal Rot for the critical reading of this manuscript, and the members of the Suzuki Laboratory for their valuable discussions. We also thank the CRUK Flow Cytometry Core Service at Barts Cancer Institute (Core Award C16420/A18066). This project was funded by the Queen Mary Innovation Proof-of-Concept fund and British Heart Foundation (Program Grants RG/15/3/31236 and RG/19/7/34577; Project Grant PG/18/77/34100). It was also supported by the UK National Institute for Health Research Biomedical Research Center at Barts.

## Author contributions

T.I. and K.S. primarily conceived the project and designed the experiments. T.I. performed most experiments with professional supports from Y.S., L.F., M.P., M.S., M.F., S.K., K.Y., K.K., and F.L. T.I and K.S. conducted the analysis and interpretation of data. K.S., T.I., M.P., and F.L. wrote and edited the manuscript with input and approval from all other authors.

## Competing interests

The authors declare no competing interests.
