## [Peer Review File · Nature Communications]

REVIEWER COMMENTS

Reviewer #1 (Remarks to the Author):

This study describes interesting experiments to evaluate the role of subsets of peritoneal macrophages in prevention of post-operative adhesions in the peritoneal cavity. Local depletion and adoptive transfer experiments reveal that resident peritoneal macrophages prevent adhesion formation. Furthermore, immunofluorescent and wholemount imaging, proximity ligation assay and blockade of CD11b convincingly demonstrate that macrophages use CD11b to bind exposed fibrin clots that form on disrupted and denuded mesothelium. As this process neither affects the degree of fibrin deposition nor mesothelial disruption, the authors present a convincing argument that peritoneal macrophages prevent adhesions by forming a physical barrier between fibrin clots and healthy mesothelium. Furthermore, the authors show that resident-like F4/80hi peritoneal macrophages are more well-adapted to perform this function than another population of peritoneal macrophages, so-called small peritoneal recruited F4/80lo macrophages, and providing much needed evidence of the diversification of function between these populations. From a therapeutic perspective, the authors also show that treatment with IL-4 complex dramatically reduces adhesion formation, which is at least in part attributable to an increase in resident macrophage numbers.

Post-operative adhesions are a common and potentially debilitating complication with few viable therapeutic options beyond corrective surgery, and hence this manuscript addresses an important unmet clinical need. The manuscript is well written and accessible to a broad readership. The rationale for the work is strong and the case is well made that understanding the role and function of macrophage subsets in post-operative adhesions is essential to inform development of macrophage-directed clinical interventions.

However I do have some concerns.

1) A major premise of the study is that heterogeneity of macrophages may explain the inconsistencies in findings between studies. However, the main contradiction remains unresolved. ie, why does pre-treatment of rabbits with proteose peptone (which would be expected to drive recruitment of large numbers of monocyte-derived macrophages and the likely loss of the tissue resident population) protect against adhesions?

Hence, while the conclusion that resident peritoneal macrophages are anti-adhesion seems robust (based on adoptive transfer and depletion data), it remains unclear whether F4/80lo macrophages recruited by surgery/ischemic injury could provide this protective function but are normally just too few in number. Hence, is it largely just a numbers game? Although adoptive transfer of CD206+ small peritoneal macrophages expanded by IL-4 treatment did not prevent adhesion formation, these cells are potentially of a unique phenotype compared with most inflammatory macrophages recruited during peritoneal inflammation (see point 2). Hence, at the very least the authors should discuss this caveat, but ideally it would be very informative to show whether adoptive transfer of large numbers of inflammation-elicited inflammatory macrophages (eg thioglycolate or similar) protect against adhesion, as this could be important both from a therapeutic perspective as well as unifying the literature.

2) Referring to CD206+ F4/80lo macrophages present following surgery as small peritoneal macrophages (SPM) is potentially inaccurate. SPM appear to have a unique transcriptional fingerprint that differs to the majority of monocyte-derived macrophages recruited during peritoneal inflammation (PMID: 23974197; PMID:27378515). As the authors only use expression of CD206 to define these cells rather than more discriminative markers such as DNAM-1 (PMID:27378515), it would be better simply to refer to them as CD206+ F4/80lo macrophages.

3) On a similar issue, the characterisation of the peritoneal macrophages is relative rudimentary,

focusing on two populations defined by CD206 and F4/80. As shown in Fig. 3 and SFig. 4, there appears to be a large number of F4/80^{lo} CD206⁻ cells present by day 3-5 that are numerically dominant over F4/80^{hi} and CD206⁺ cells and could represent monocytes/CD206⁻ recruited macrophages but which have been ignored. These cells warrant further characterisation (particularly as they could contribute to F4/80⁺ CD206⁻ cells detected on the fibrin clots by immunofluorescence in SFig. 3). Although the liposome up-take assay suggests they are not phagocytic, the timepoint at which this analysis was performed is not clear. Re-analysis of CCR2 and CD11b expression on these cells is warranted, but also additional staining for macrophage/monocyte-lineage markers CSF1R and Ly6C and granulocyte lineage markers Ly6G/SiglecF would help to refine whether these are monocytes/macrophages or granulocytes. Similarly, using additional markers for immunofluorescence imaging that distinguish resident and recruited macrophages would strengthen the conclusion that only resident macrophages bind fibrin clots.

Minor comments:

- It is not clear what the benefits of the surgical ligation model are over other models of peritoneal adhesions. Given that the role of macrophages in adhesion formation appears model dependent, the ms would benefit from an indication of why the button model was chosen over mechanical scraping/implantation of gauze etc, and whether/how it is physiologically superior.
- Line 283, when discussing the source of cells that reconstitute the resident peritoneal population following surgery, the authors should reference recent findings that laparotomy alone causes partial replacement of resident LPM by BM-derived cells (PMID: 32561560).
- Fig 6A: the gate used to identify CD206⁺ SPM in PBS control mice appears to exclude most of these cells in control mice as the cut-off for F4/80 is set too high.

Reviewer #2 (Remarks to the Author):

In this study the authors use a murine model of adhesions and found that resident F4/80^{High}CD206⁻ peritoneal macrophages accumulated and formed a 'macrophage barrier' to shield the fibrin clots in place of the lost mesothelium. They identify that depletion of this macrophage subset or blockage of CD11b impaired the macrophage barrier and exacerbated adhesions. augmented by interleukin-4-based treatment or adoptive transfer of this macrophage subset, resulting in prevention of adhesions in this murine model. The paper is well written and the experiments, although mostly observational, were reasonable. There is a lack of mechanistic insight though, as to why IL4 is augmenting this process and it is very surprising, given the pleiotropic effects of IL4, that it does not affect T cells or mast cells/histamine, etc. that are well documented effects of IL4. I have the following specific comments:

1. Macrophages are often identified in tissue based on their functions – phagocytosis/killing capacity, cytokine/mediator production. This should be examined as part of Figure 2, instead of using surface markers alone for identification of macrophages.
2. It is surprising that following surgical injury, recruited monocytes would not be more involved. It seems that cell labeling/tracking experiments and/or labeled adoptive transfer experiments should be done to determine the contribution of the recruited monocytes following injury – surely they are playing some role post-injury.
3. IL4 is pleiotropic – no mechanism is explored as to why/how this alters the macrophages – and

not other cells in the area – especially T cells/eosinophils. What are the effects of IL4 blockade on adhesions.

4. The translational potential of this work done in a single murine model is unclear – use of human cells and/or a second model would improve the potential findings from this study to make this potentially more broadly applicable.

We thank the Editor and Reviewers for the careful review of our manuscript. We are pleased that the novelty and impact of our manuscript are appreciated. According to the constructive comments provided, we have improved our manuscript by adding new experimental data and altering the texts and figures, so that we have now addressed all issues raised. Please find our responses to the individual points as follows.

Point-by-point response to the referees' comments

Reviewer comments are in black, and our responses are in blue.

Reviewer #1:

This study describes interesting experiments to evaluate the role of subsets of peritoneal macrophages in prevention of post-operative adhesions in the peritoneal cavity. Local depletion and adoptive transfer experiments reveal that resident peritoneal macrophages prevent adhesion formation. Furthermore, immunofluorescent and wholemount imaging, proximity ligation assay and blockade of CD11b convincingly demonstrate that macrophages use CD11b to bind exposed fibrin clots that form on disrupted and denuded mesothelium. As this process neither affects the degree of fibrin deposition nor mesothelial disruption, the authors present a convincing argument that peritoneal macrophages prevent adhesions by forming a physical barrier between fibrin clots and healthy mesothelium. Furthermore, the authors show that resident-like F4/80^{hi} peritoneal macrophages are more well-adapted to perform this function than another population of peritoneal macrophages, so-called small peritoneal recruited F4/80^{lo} macrophages, and providing much need evidence of the diversification of function between these populations. From a therapeutic perspective, the authors also show that treatment with IL-4 complex dramatically reduces adhesion formation, which is at least in part attributable to an increase in resident macrophage numbers. Post-operative adhesions are a common and potentially debilitating complication with few viable therapeutic options beyond corrective surgery, and hence this manuscript addresses an important unmet clinical need. The manuscript is well written and accessible to a broad readership. The rationale for the work is strong and the case is well made that understanding the role and function of macrophage subsets in post-operative adhesions is essential to inform development of macrophage-directed clinical interventions. However I do have some concerns.

(1) A major premise of the study is that heterogeneity of macrophages may explain the inconsistencies in findings between studies. However, the main contradiction remains unresolved. ie, why does pre-treatment of rabbits with proteose peptone (which would be expected to drive recruitment of large numbers of monocyte-derived macrophages and the likely loss of the tissue resident population) protect against adhesions?

Hence, while the conclusion that resident peritoneal macrophages are anti-adhesion seems robust (based on adoptive transfer and depletion data), it remains unclear whether F4/80^{lo} macrophages recruited by surgery/ischemic injury could provide this protective function but are normally just too few in number. Hence, is it largely just a numbers game? Although adoptive transfer of CD206⁺ small peritoneal macrophages expanded by IL-4 treatment did not prevent adhesion formation, these cells are potentially of a unique phenotype compared with most inflammatory macrophages recruited during peritoneal inflammation (see point 2). Hence, at the very least the authors should discuss this caveat, but ideally it would be very informative to show whether adoptive transfer of large numbers of inflammation-elicited inflammatory macrophages (eg thioglycolate or similar) protect against adhesion, as this could be important both from a therapeutic perspective as well as unifying the literature.

We thank the Reviewer for this insightful comment. Following the recommendation, we performed additional studies to further characterise the role of recruited inflammatory monocytes/macrophages. Flow cytometry analysis showed that the F4/80^{Low}CD206⁻ cell fraction in the peritoneal fluid contained a significant number of CD11b⁺CCR2⁺Ly6C^{High} classical (inflammatory) monocytes/macrophages at day 3 post-surgery (New Supplementary Fig. 8a, b). However, additional immunolabelling demonstrated that the accumulated cells on the button surface were negative for CCR2, indicating that recruited inflammatory monocytes/macrophages did not contribute to the formation of anti-adhesion cell barriers (New Supplementary Fig. 9a). This inability of recruited monocytes/macrophages to participate in the cell barrier formation was further confirmed in the “repeated ischemic button creation model” (New Supplementary Fig. 10a, b) instead of adoptive transfer suggested by the Reviewer. In this supplementary study, we created an additional ischemic button on the right-side peritoneal wall at day 3 after the first ischemic button creation on the left-side peritoneal wall. This model enabled to investigate the role of recruited monocytes/macrophages more precisely because the occurrence of these recruited cells in the peritoneal cavity is much more abundant at day 3 compared to the time period for the ordinary cell barrier formation (within hours post-surgery, in which the presence of these

recruited cells is limited). The obtained result demonstrated that even such an increased occurrence did not allow these cells to contribute to the cell barrier formation. It is therefore confirmed that recruited monocytes/macrophages do not have an ability to form an anti-adhesion cell barrier, and that the cell barrier formation is not a simple “numbers game”; only resident peritoneal macrophages have this function.

Aforementioned results do not necessarily contradict to the previous report showing that inflammation-elicited inflammatory macrophages protect against adhesion. This is because these cells may be able to activate a different anti-adhesion mechanism(s), potentially including modulation of inflammation, fibrin formation, fibrinolysis and/or vacuolisation and fibrosis of adhesion tissues. It will be of interest to investigate the detailed role of inflammation-elicited inflammatory macrophages; however, we could not afford to perform such an experiment because it will not add critically important information to our manuscript that specifically focuses on the anti-adhesion macrophage barrier. Further research is warranted to this end. We have added this discussion in the revised manuscript (page 13, line 324-337).

(2) Referring to CD206⁺ F4/80^{lo} macrophages present following surgery as small peritoneal macrophages (SPM) is potentially inaccurate. SPM appear to have a unique transcriptional fingerprint that differs to the majority of monocyte-derived macrophages recruited during peritoneal inflammation (PMID: 23974197; PMID:27378515). As the authors only use expression of CD206 to define these cells rather than more discriminative markers such as DNAM-1 (PMID:27378515), it would be better simply to refer to them as CD206⁺ F4/80^{lo} macrophages.

We appreciate this thoughtful suggestion. We agree that F4/80^{Low}CD206⁺ macrophages in our manuscript are not identical to previously reported small peritoneal macrophages (SPM). Therefore, as recommended by the Reviewer, we now refer to these cells as F4/80^{Low}CD206⁺ macrophages throughout our revised manuscript and have discussed this macrophage subset in the discussion (page 12-13, line 308-323).

(3) On a similar issue, the characterisation of the peritoneal macrophages is relative rudimentary, focusing on two populations defined by CD206 and F4/80. As shown in Fig. 3 and SFig. 4, there appears to be a large number of F4/80^{lo} CD206⁻ cells present by day 3-5 that are numerically dominant over F4/80^{hi} and CD206⁺ cells and could represent

monocytes/CD206⁻ recruited macrophages but which have been ignored. These cells warrant further characterisation (particularly as they could contribute to F4/80⁺ CD206⁻ cells detected on the fibrin clots by immunofluorescence in SFig. 3). Although the liposome uptake assay suggests they are not phagocytic, the timepoint at which this analysis was performed is not clear. Re-analysis of CCR2 and CD11b expression on these cells is warranted, but also additional staining for macrophage/monocyte-lineage markers CSF1R and Ly6C and granulocyte lineage markers Ly6G/SiglecF would help to refine whether these are monocytes/macrophages or granulocytes. Similarly, using additional markers for immunofluorescence imaging that distinguish resident and recruited macrophages would strengthen the conclusion that only resident macrophages bind fibrin clots.

We thank the Reviewer for this constructive comment. We performed all recommended experiments, and the results obtained have fully addressed the concerns raised. We characterized F4/80^{Low}CD206⁻ cells by using flow cytometry for CCR2, CD11b, CSF1R, Ly6C, Ly6G and Siglec-F. As a result it was shown that F4/80^{Low}CD206⁻ cells were composed of Ly6C^{High} classical monocytes/macrophages ($22.8 \pm 4.6\%$), Ly6C^{Low} non-classical monocytes/macrophages ($7.5 \pm 1.1\%$) and Siglec-F⁺ eosinophils ($53.2 \pm 4.2\%$), while Ly6G⁺ neutrophils were scarcely found in this population (New Supplementary Fig. 8a, b). In addition, we re-analysed the data of liposome uptake assay, which demonstrated that F4/80^{Low}CD206⁻ cells collected at day 3 or day 5 post-surgery did not have a phagocytic ability, suggesting that these cells are unlikely to be mature macrophages (New Fig. 3c, d). We also performed additional immunostaining studies, which described that CCR2, SiglecF or Ly6G positive cells were seldom found on the button surface (New Supplementary Fig. 9a), and that F4/80⁺ cells accumulated on the button surface were positive for a resident macrophage marker, ICAM2 (New Supplementary Fig. 9b, c). In addition, the data from the “repeated ischemic button creation model” (please refer to our response to the issue (1) above) further confirmed that CCR2, Ly6G or SiglecF positive cells were not involved in the cell barriers (New Supplementary Fig. 10b). These supplementary data collectively strengthen our finding that resident macrophages, but not recruited macrophages or other cell types, have a function to form an anti-adhesion barrier. These new data, statements and additional methodological information have been added in New Fig. 3c, d, New Supplementary Fig. 8-10 and text (page 7-8, line 161-181).

Minor comments:

- It is not clear what the benefits of the surgical ligation model are over other models of peritoneal adhesions. Given that the role of macrophages in adhesion formation appears model dependent, the ms would benefit from an indication of why the button model was chosen over mechanical scraping/implantation of gauze etc, and whether/how it is physiologically superior.

We agree that an appropriate choice of the model is critical in research of post-operative peritoneal adhesions. As previously reported (PMID: 20304431), the ischemic button model has an important advantage in inducing a reproducible and adequate degree of peritoneal adhesions compared to other models. We have also confirmed this superiority in our previous pilot study. Scraping/abrasion of the peritoneal wall or cecum is one of the frequently used methods; however, we found it technically difficult to produce a consistent depth and/or width of mechanical damage by scraping. In addition, a varying degree of bleeding was seen. As a result, the degree of induced adhesions was not reproducible sufficiently. Implantation of a gauze is another option, while the implanted gauze was unexpectedly movable in the peritoneal cavity due to peristalsis of the intestines, resulting in variable formation of peritoneal adhesions. In addition, an implanted gauze could cause extra inflammation as a result of foreign body reaction. In response to the Reviewer's comment, we have added this statement in our revised manuscript (page 14, line 370-372).

- Line 283, when discussing the source of cells that reconstitute the resident peritoneal population following surgery, the authors should reference recent findings that laparotomy alone causes partial replacement of resident LPM by BM-derived cells (PMID: 32561560).

Thank you very much for this valuable suggestion. We have now cited and discussed this paper in our manuscript (page 12, line 307).

- Fig 6A: the gate used to identify CD206⁺ SPM in PBS control mice appears to exclude most of these cells in control mice as the cut-off for F4/80 is set too high.

Thank you very much for the concern. We have set this "high" F4/80 cut-off intentionally. As F4/80^{-(or very low)}CD206⁺ cells are unlikely to be mature macrophages judging from their low phagocytosis capacity (New Fig. 3c, d), we would like to exclude these cells from the F4/80^{Low}CD206⁺ macrophage population that exhibits a robust phagocytosis function.

Reviewer #2:

In this study the authors use a murine model of adhesions and found that resident F4/80^{High}CD206⁻ peritoneal macrophages accumulated and formed a ‘macrophage barrier’ to shield the fibrin clots in place of the lost mesothelium. They identify that depletion of this macrophage subset or blockage of CD11b impaired the macrophage barrier and exacerbated adhesions. augmented by interleukin-4-based treatment or adoptive transfer of this macrophage subset, resulting in prevention of adhesions in this murine model. The paper is well written and the experiments, although mostly observational, were reasonable. There is a lack of mechanistic insight though, as to why IL4 is augmenting this process and it is very surprising, given the pleiotropic effects of IL4, that it does not affect T cells or mast cells/histamine, etc. that are well documented effects of IL4. I have the following specific comments:

1. Macrophages are often identified in tissue based on their functions – phagocytosis/killing capacity, cytokine/mediator production. This should be examined as part of Figure 2, instead of using surface markers alone for identification of macrophages.

We would like to thank the Reviewer for this constructive comment. As recommended, we have now added the data on the functional characterisation of macrophages. Firstly, we refined the phagocytosis analysis, which now presents the distinct liposome uptake ability among macrophage/monocyte subpopulations (New Fig. 3c, d). We also added histological evidence showing that F4/80⁺ macrophages accumulated on the ischemic button surface have a phagocytosis ability (New Supplementary Fig. 3a, b). Secondly, we assessed the mRNA expression levels of major cytokine/mediator genes in the macrophage subpopulations (New Supplementary Fig. 6a, b). Expression of *Tnfa* and *Il1b* genes was low in F4/80^{High}CD206⁻ peritoneal macrophages that contribute to anti-adhesion cell barriers, suggesting that this macrophage subset is unlikely to be pro-inflammatory.

2. It is surprising that following surgical injury, recruited monocytes would not be more involved. It seems that cell labeling/tracking experiments and/or labeled adoptive transfer experiments should be done to determine the contribution of the recruited monocytes following injury – surely they are playing some role post-injury.

In response to this thoughtful comment, we performed additional investigations regarding recruited monocytes following injury. We first analysed peritoneal cavity cells at day 3 post-

surgery and found that sizeable numbers of CD11b⁺CCR2⁺Ly6C^{High} classical monocytes/macrophages and CD11b⁺CCR2⁺Ly6C^{Low} non-classical monocytes/macrophages were included in the F4/80^{Low}CD206⁻ cell population (New Supplementary Fig. 8b). Liposome uptake assay demonstrated that these F4/80^{Low}CD206⁻ cells did not have a phagocytic ability, suggesting that they are unlikely to be mature macrophages (New Fig. 3c, d). Furthermore, additional immunolabelling analysis clarified that accumulated cells on the button surface were negative for CCR2, indicating that above-mentioned recruited monocytes/macrophages did not participate in the formation of anti-adhesion cell barriers (New Supplementary Fig. 9a). Such inability of recruited monocytes to form a cell barrier was further confirmed in the “repeated ischemic button creation model” (New Supplementary Fig. 10a, b). In this model, we created an additional ischemic button on the left-side peritoneal wall at day 3 after the first ischemic button creation on the right-side peritoneal wall. This model enabled to investigate the role of recruited monocytes/macrophages more precisely because the occurrence of these recruited cells in the peritoneal cavity is more abundant at day 3 compared to the time of the ordinary cell barrier formation (within hours post-surgery, in which only a limited number of recruited monocytes are present). We confirmed that, even in this model, recruited monocytes/macrophages did not contribute to the cell barrier formation.

Taken together, our results certainly indicate that recruited monocytes do not participate in the anti-adhesion barrier formation. Having said this, we agree with the Reviewer that recruited monocytes may play a role in adhesion formation post-surgery; however, this is likely through a different mechanism(s), e.g. modulation of inflammation, fibrin formation and/or fibrinolysis. Although a further comprehensive, systematic study on this cell type may be interesting, we could not afford to perform it in this occasion because it will not add critically important information to our manuscript that focuses on the macrophage barrier. Future studies specifically investigate the role of recruited monocytes are warranted. We have added these discussions in the revised manuscript (page 13, line 324-337).

3. IL4 is pleiotropic – no mechanism is explored as to why/how this alters the macrophages – and not other cells in the area – especially T cells/eosinophils. What are the effects of IL4 blockade on adhesions.

Thank you very much for this insightful comment. It has been robustly demonstrated that IL-4 administration drives proliferation of tissue-resident macrophages (PMID: 21566158,

PMID: 24101381). Consistent with these reports, we observed that IL-4c administration markedly increased the number of F4/80^{High}CD206⁻ resident peritoneal macrophages (Fig. 6a, b). The adoptive transfer study demonstrated that these F4/80^{High}CD206⁻ resident peritoneal macrophages have a vigorous ability to attenuate post-operative adhesion formation (Fig. 6j-k). In addition, IL-4c administration increased the number of F4/80^{Low}CD206⁺ recruited macrophages, while adoptive transfer of these macrophages did not reduce adhesion formation significantly.

By contrast, the flow cytometry and immunohistolabeling studies demonstrated that the number of B220⁺ B lymphocytes, CD3⁺CD4⁺ T lymphocytes, CD3⁺CD8⁺ T lymphocytes or Ly6G⁺ neutrophils in the peritoneal fluid or on/in the ischemic button was not affected by IL-4c administration in our model (Previous Supplementary Fig. 8; New Supplementary Fig. 13). To strengthen this observation, we added new data showing the profile of 111 soluble protein levels in the peritoneal fluid at day 3 post-surgery as measured by using antibody array (New Supplementary Fig. 14). This revealed that the major Th1/Th2-related cytokines were expressed at a very low level in the peritoneal fluid, and that this insignificant expression was not altered by IL4c administration. On the other hand, a macrophage-derived chemokine, CCL6/C10, was up-regulated by the IL-4c treatment, further supporting the active effect of IL-4c on macrophages.

Based on these data, we confirm that our protocol of IL-4c administration increases the number of macrophages, but not other cell types, in our mouse model of ischemic button creation. However, we cannot exclude a possibility of functional alterations of non-macrophage cell types in response to IL-4, which may modulate a process of adhesion formation. Further detailed cell-type dependent responses to IL-4 and the resulting effects on adhesion formation could be investigated by using cell-type specific IL-4 receptor knockout mice. We have added this statement as a limitation of our study in the revised manuscript (page 13-14, line 338-351).

As regards the IL-4 blockage study suggested by the Reviewer, it has already been reported that the post-operative adhesion is not affected in IL-4 knockout mice compared to wild-type mice (PMID: 26313905), suggesting that endogenous IL-4 is unlikely to play a critical role in the formation of adhesions. This finding is reasonable, considering our data that the level of endogenous IL-4 in the peritoneal cavity is low (New Supplementary Fig. 14). We have added this discussion in the revised manuscript (page 13-14, line 338-351).

4. The translational potential of this work done in a single murine model is unclear – use of human cells and/or a second model would improve the potential findings from this study to make this potentially more broadly applicable.

We appreciate this comment useful for future expansion of this study. Our study utilised the ischemic button model, which has been demonstrated to have an important advantage in inducing consistent and reproducible intra-abdominal adhesions compared to other models (PMID: 20304431). For clinical translation of the results obtained in this study, it will be needed to confirm whether these findings are true when human cells or a different model, particularly a large animal model, are used. We are afraid, however, that these studies are beyond the focus of our present basic science research and may be planned in the subsequent translational development. We have added this discussion in the revised manuscript (page 14, line 370-375).

REVIEWERS' COMMENTS

Reviewer #1 (Remarks to the Author):

Thank you for addressing my comments

Reviewer #2 (Remarks to the Author):

The authors have satisfactorily answered my critiques.